# Saponins of Selected Triterpenoids as Potential Therapeutic Agents: A Review

**DOI:** 10.3390/ph16030386

**Published:** 2023-03-02

**Authors:** Uladzimir Bildziukevich, Martina Wimmerová, Zdeněk Wimmer

**Affiliations:** 1Isotope Laboratory, Institute of Experimental Botany AS CR, Vídeňská 1083, 14220 Prague 4, Czech Republic; 2Department of Chemistry of Natural Compounds, University of Chemistry and Technology in Prague, Technická 5, 16628 Prague 6, Czech Republic

**Keywords:** oleanane, ursane, lupane, saponin, glycoconjugate, pharmacological effects

## Abstract

Saponins represent important natural derivatives of plant triterpenoids that are secondary plant metabolites. Saponins, also named glycoconjugates, are available both as natural and synthetic products. This review is focused on saponins of the oleanane, ursane, and lupane types of triterpenoids that include several plant triterpenoids displaying various important pharmacological effects. Additional convenient structural modifications of naturally-occurring plant products often result in enhancing the pharmacological effects of the parent natural structures. This is an important objective for all semisynthetic modifications of the reviewed plant products, and it is included in this review paper as well. The period covered by this review (2019–2022) is relatively short, mainly due to the existence of previously published review papers in recent years.

## 1. Introduction

Pentacyclic triterpenoids are among the most important plant products, even if they are still less pharmacologically explored. Among these secondary plant metabolites, the most frequently studied triterpenoids are those with oleanane, ursane and lupane skeletons, represented mainly by oleanolic acid (**1a**), ursolic acid (**1b**), and betulinic acid (**1c**) (Figure 1). Other pentacyclic triterpenoids less frequently occurring in nature, namely glycyrrhizic acid (**1d**), glycyrrhetinic acid (**1e**), quillaic acid (**1f**), and echinocystic acid (**1g**) (Figure 1), are among the most frequent oleanane-type of triterpenoids, and several even less frequent structures are also included into the respective parts of this review paper. All these triterpenoids display important therapeutic potential [1,2,3,4,5,6,7]. However, a general disadvantage of plant triterpenoids consists in their low solubility in water and aqueous media, and limited bioavailability. This disadvantage can be solved by designing and preparing suitable derivatives with enhanced pharmacological activity and bioavailability in comparison with the parent plant triterpenoids, as reviewed recently [3]. In natural resources, all triterpenoids mostly appear as aglycones in triterpenoid saponins that may be considered glycoconjugates [8,9,10,11,12].

The olive tree (*Olea europaea* L.; Oleaceae) is the most important plant source of oleanolic acid, (3β)-3-hydroxy-olean-12-en-28-oic acid (**1a**), found mainly in its leaves and fruits [8,13]. However, olive leaves, bark, and fruits also contain a small amount of ursolic acid (**1b**) [14]. In natural resources, both of these triterpenoids often appear simultaneously. The name of oleanolic acid (**1a**) is based on the name of the olive plant, an important food, nutritive, and medicinal species. Other natural resources of oleanolic acid (**1a**) are mostly medicinal plants, of which the most important ones are *Panax ginseng* C.A.Mey (Araliaceae; namely the root), *Arctostaphyllos uva-ursi* L. Spreng. (Ericaceae; bearberry), *Calluna vulgaris* L. Hull (Ericaceae; heather), *Crataeva nurvala* Buch. Ham. (Capparaceae; three-leaved caper), *Rosmarinus officinalis* L. (Lamiaceae; rosemary), *Sambucus chinensis* Lindl. (Adoxaceae; Chinese elder), *Solanum incanum* L. (Solanaceae; Sodom’s apple), or *Syzygium aromaticum* L. Merr. & L.M. Perry (Myrtaceae; clove), and then fungi, e.g., *Ganoderma lucidum* Karst (Ganodermataceae; reishi) [15].

Ursolic acid, (3β)-3-hydroxy-urs-12-en-28-oic acid (**1b**), is often considered to be the isomer of oleanolic acid (**1a**), bearing similar structural characteristics. Ursolic acid (**1b**) has been found in many nutritive, food, or medicinal plants, e.g., apple (*Malus domestica* Borkh.; Rosaceae; fruit peel), bearberry (*Arctostaphylos uva-ursi* L. Spreng.; Ericaceae; leaves), black elder (*Sambucus nigra* L.; Adoxaceae; leaves and bark), coffee (*Coffea arabica* L.; Rubiaceae; the leaves), hawthorn (*Crataegus* L. spp.; Rosaceae; the leaves and flowers), eucalyptus (*Eucalyptus* L’Hér.; Myrtaceae; the leaves and bark), lavender (*Lavandula angustifolia* Mill.; Lamiaceae; the leaves and flowers), marjoram (*Origanum majorana* L.; Lamiaceae; leaves), oregano (*Origanum vulgare* L.; Lamiaceae; the leaves), oleander (*Nerium oleander* L.; Apocynaceae; the leaves), sage (*Salvia officinalis* L.; Lamiaceae; the leaves), or thyme (*Thymus vulgaris* L.; Lamiaceae; the leaves), and in the wax layers of many edible fruits [14,16].

Betulinic acid, (3β)-3-hydroxy-lup-20(29)-en-28-oic acid (**1c**), is the lupane-type of pentacyclic triterpenoids. It is widespread in the bark of the birch tree (*Betula* L. spp., Betulaceae). The bark contains a number of medicinally important plant products [17]. Other sources of betulinic acid (**1c**) include *Diospyros* L. spp. (Ebenaceae), *Paeonia* L. spp. (Paeoniaceae), *Platanus* L. spp. (Platanaceae), *Syzygium* P. Browne ex Gaertn. spp. (Myrtaceae), and *Ziziphus* Mill. spp. (Rhamnaceae), of which most plants belong to the medicinal species, often used in traditional medicines [18].

In general, pentacyclic triterpenoids have displayed their capability of forming nanoscale materials that have various important physicochemical characteristics [19,20,21,22]. Self-assembly was also observed with other natural products, e.g., sterols and their derivatives, including derivatives of steryl saponins [23]. Later on, different derivatives of natural triterpenoids have also been subjected to a similar investigation resulting in the findings that a number of those compounds also display the analogous ability to self-assemble, and the formed nanosized materials exhibited important physicochemical and pharmacological characteristics, making those triterpenoid-based materials attractive for a more detailed investigation of their physicochemical properties and pharmacological effects [24,25,26,27,28]. Nanosized materials of this origin often display amphiphilic properties, which are important for pharmacological and medicinal applications. Saponins that are considered to be glycoconjugates with similar characteristics as amphiphiles, both appearing in nature and/or available synthetically, representing compounds and nanoscale materials formed therefrom that are capable of giving rise to multivalent carbohydrate-mediated interactions in biological processes that include important disease mechanisms [29]. Saponins also play an important role as adjuvants, i.e., compounds or nanoscale systems capable of increasing the efficacy of certain drugs [30].

Natural triterpenoids, namely those mentioned in this review in more detail, display a broad spectra of pharmacological effects that are well documented in the literature and have been reviewed recently [3,31,32,33]. As already stated, due to the limited solubility of triterpenoids in aqueous media, they mostly appear as triterpenoid saponins in plant resources. Therefore, in the objectives of this review paper, triterpenoid saponins (glycoconjugates) were targeted, with special attention paid to those bearing oleanane, ursane, and lupane skeletons of aglycones. Because several review papers have been recently published on the synthesis of triterpenoid saponins (glycoconjugates), including the evaluation of their pharmacological effects [11,12,34], the period has been selected in this review paper to cover original papers appearing even more recently (2019–2022). Triterpenoid saponins were generally reported to display anticancer, antiviral, antibacterial, anti-inflammatory, anti-Alzheimer, antileishmanial, antioxidant, and immunomodulatory effects, and the ability to inhibit α-glucosidase [34]. They represent an important group of natural compounds, available also by chemical synthesis (glycoconjugates). Even if several review papers have been published on this topic in the period 2015–2021 [5,6,7,11,12,31,32,33,34], the period of the most recent four years, 2019–2022, has not been fully covered. Therefore, attention has been focused on the most recent findings that have appeared in the literature during several of the most recent years, and highlights original papers that appeared in the period not being reviewed before. The original approach of this review paper is that (a) it has covered and highlighted the most recent four years of investigation in the field, and (b) it has been focused on saponins of triterpenoids arranged according to the structure of the triterpenoid aglycone, with the comments and evaluation of the pharmacological activity, mode of action, and physicochemical characteristics, whenever those items of information have been available in the original papers included in this review article.

## 2. Triterpenoids with the Oleanane Skeleton

Oleanane-type triterpenoids are the most widespread plant triterpenoids from the reviewed types of triterpenoids. They include not only the intensively studied oleanolic acid (**1a**), but also glycyrrhizic (**1d**), glycyrrhetinic (**1e**), quillaic (**1f**) and echinocystic acid (**1g**) (Figure 1), and several other oleanane-type triterpenoid saponins, e.g., monellosides, etc., that are mentioned in this review paper in the respective paragraphs below.

Five oleanane-type triterpenoid saponins (**2a**–**2e**; Figure 2) were isolated from the leaves of *Aralia dasyphylla* Miq. (Araliaceae), besides ursane-type triterpenoid saponins that are mentioned below in the text [35]. All isolated compounds were evaluated in vitro for their cytotoxicity in three human cancer cell lines, i.e., human hepatocellular carcinoma (HepG2), human lung adenocarcinoma (LU-1), and human rhabdomyosarcoma (RD), and in silico by molecular docking studies on human glucose transporter 1 (hGLUT1) protein. The triterpenoids **2a**, **2c,** and **2d** exhibited good growth inhibition of HepG2 and LU-1 cancer cell lines with IC_50_ values in the range of 1.76–7.21 µM (Table 1). The triterpenoid **2c** was the compound with the highest cytotoxicity of this series of compounds, capable of inhibiting all tested cancer cell lines with IC_50_ values of 2.73 ± 0.12 µM (HepG2), 1.76 ± 0.11 µM (LU-1), and 2.63 ± 0.10 µM (RD), respectively (Table 1). The in silico calculations of absorption, distribution, metabolism, excretion, and oral toxicity (ADMET) parameters, and molecular docking study results with **2a**–**2e** showed that compound **2a** had one of the highest binding affinities to hGLUT1 [35]. Therefore, the presented results focused attention on developing potential hGLUT1 inhibitors elatoside E (**2a**), 3-*O*-[β-d-glucopyranosyl(l→3)]-α-l-arabinopyranosyl oleanolic acid (**2c**), 3-*O*-α-l-arabinopyranosyl oleanolic acid (**2d**), and oleanolic acid 28-*O*-β-d-glucopyranosyl ester (**2e**) that were evaluated to become worthy of further investigation for the prevention or treatment of diabetes and cancer (Table 1). Based on other calculated data, the intestinal absorption increased in the series **2a** < **2c** < **2d** < **2e**.

Other oleanane-type saponins **3a** and **3b** (Figure 3) were isolated from plants of the genus *Weigela* Thunb. (Caprifoliaceae) [36] that is composed of about ten species, mainly distributed in Asia, but can be found in Europe as well [37]. Over 200 cultivars have been produced for ornamental purposes [36,37]. The oleanane saponins were generally reported to possess a spectrum of pharmacological activities that include anti-inflammatory [38], anticomplementary [39], stimulatory [40] and cytotoxic activity [37,41,42,43]. They are also capable of recognizing antibodies [44]. A detailed study with **3a** and **3b** has not yet been published [36]; however, based on the investigation performed by other authors [45], it is supposed that the important pharmacological potential of **3a** and **3b** will be evaluated later on.

*Weigela* species were also studied by other authors [45], who isolated additional oleanane-type saponins (**4a**–**4d**; Figure 4 and **5a**–**5f**; Figure 5) from three *Weigela* (Thunb.) hybrids and cultivars: *W. x Styriaca*, *W. florida* “Minor black”, and *W. florida* “Brigela” [45]. The authors elucidated the structures of all new compounds of the series of **4a**–**4d** (Figure 4) and **5a**–**5f** (Figure 5) [45]. The tested saponins **4a**–**4d** and **5a**–**5f** showed antifungal activity (*Candida albicans*) and antibacterial activity (*Staphylococcus aureus*, *Pseudomonas aeruginosa*) with the minimum inhibitory concentrations MIC ~200 µg · mL^−1^ [45].

A series of 13 so far undescribed triterpenoid saponins, named monellosides A-M (**6a**–**6m**; Figure 6), were isolated from the aerial parts of *Anagallis monelli* ssp. *linifolia* (L.) Maire (Primulaceae), together with a series of ten already known oleanane-type glycosides [46]. The structures of the isolated compounds were elucidated by the relevant analytical methods (1D and 2D-NMR spectroscopy and HR-ESI-MS), followed by acid hydrolysis to liberate the triterpenoid aglycones. Monellosides A-M (**6a**–**6m**; Figure 6) have a carbohydrate chain linked on the C(3)-OH of the aglycone with a common β-d-glucopyranosyl-(1→4)-α-l-arabinopyranosyl sequence, which was further glycosylated by an additional glucose and/or xylose unit. The sequence of β-d-xylopyranosyl-(1→2)-β-d-glucopyranosyl-(1→4)-[β-d-glucopyranosyl-(1→2)-]α-l-arabinopyranosyl oligosaccharide motif was common to all the 13,28-epoxyoleanane core skeletons except for the compounds **6j**–**6m** (Figure 6) of the series. It seems to be a general feature of the saponins from the species of Myrsinaceae and Primulaceae families that they bear the 13,28-epoxy-bridged skeleton of the pentacyclic triterpenoids found therein. The finding of the phytochemical results contributed to increasing the knowledge of the structures of the saponins of the genus *Anagallis* and their chemotaxonomy, and stimulated further evaluation of the biological effects of these saponins [46]. The pharmacological potential of **6a**–**6m** (Figure 6) has not yet been fully evaluated.

Three new oleanane-type triterpenoid saponins (**7a**–**7c**; Figure 7) were isolated and identified in the extracts from the stem bark of mertajam, *Lepisanthes rubiginosa* Roxb. (Sapindaceae), a plant growing in Asia, namely in Malaysia, Thailand, and Vietnam, and in the tropical areas of Africa and north-western Australia [47]. The plant has been extensively used in traditional medicine, namely to treat fever, headache, cough, diarrhea, dysentery, and jaundice [48]. Cytotoxic, antibacterial, antifungal, antioxidant, antihyperglycemic, and antipruritic activity had been proven for the extracts of *L. rubiginosa* [49]. The plant is a rich natural source of pentacyclic triterpenoids and their saponins, flavonoids, phenolic acids, tannins, and other phenolic products [49]. Finally, an intensive investigation resulted in the isolation of several new triterpenoid glycosides, lepiginosides (**7a**–**7c**) [48]. The potential antibacterial activity of the isolated compounds was tested in *Klebsiella pneumoniae*, *Pseudomonas aeruginosa*, *Salmonella typhimurium*, *Shigella sonnei*, *Escherichia coli*, *Staphylococcus haemolyticus*, and in the methicillin-resistant *Staphylococcus aureus*; however, none of the investigated triterpenoid saponins showed significant activity [47]. Therefore, it seems reasonable that a synergic action of different constituents in the extracts of *L. rubiginosa* is required for the pharmacological effects to be displayed, as it was found for the plant extracts.

Xanthoceraside (**8**; Figure 8), another oleanane-type saponin, was isolated from *Xanthoceras sorbifolium* Bunge (Sapindaceae; yellowhorn), a plant widely used in traditional medicine in China and Russia [50]. The compound has been proven to be a potent agent for treating major depressive disorders [50], namely in connection with chronic corticosterone administration. Corticosterone administration induced anxiety and depression-like abnormal behavior, caused a decrease in the expression levels of the brain-derived neurotrophic factor and a decrease in the phosphorylation of protein kinase B (AKT), the mammalian target of rapamycin (mTOR), and, finally, a decrease in a cAMP response element binding protein (CREB) in the prefrontal cortex (PFC). Therefore, xanthoceraside (**8**; Figure 8) became an attractive candidate for pharmacotherapy to treat major depressive disorder (MDD) with hypothalamic–pituitary–adrenal (HPA) axis dysfunction.

The chronic corticosterone (CORT) administration model was employed in the investigation of the effects of the studied compound, because previous investigations resulted in reports suggesting the relationship between CORT and brain-derived neurotrophic factor (BDNF) expression [51]. Mice that were chronically administered with CORT (20 mg · kg^−1^ per day) for three weeks enabled the investigation conducted to find whether chronic CORT administration impairs emotional function in mice by using the open field, social interaction, and novelty-suppressed feeding tests. Finally, the results suggested that chronic CORT administration induced anxiety-like and depression-like abnormal behavior in mice [50], and xanthoceraside (**8**; Figure 8) was capable of treating such behavior based on depressive disorders.

Platycosides having the general structures **9a**–**9c** shown in Figure 9, another series of the saponins of the oleanane-type of triterpenoids, have been isolated and identified from *Platycodon grandiflorus* Jacq. (Campanulaceae; Chinese bellflower), a well-known edible and medicinal plant [52]. The general formula **9a** represents the platycodigenin-type of compounds, **9b** represents the platygalacic-acid-type of compounds, and, finally, **9c** represents the platyconic acid-type of compounds. Platycosides comprise 56 described plant products altogether [52]. They are the main active constituents of *P. grandiflorus,* with multiple pharmacological activities. The substituents R_1_ and R_2_ in **9a**–**9c** represent a combination of monosaccharide-based and oligosaccharide-based substituents, or a hydrogen atom [52]. *P. grandiflorus* was used as a dietary supplement and functional food for relieving pulmonary disorders [52]. The investigation was focused on the metabolism of the identified platycosides in vivo, because the mechanism of platycosides had not been fully clarified. The study resulted in an important finding that 3-*O*-β-d-glucopyranosyl platycosides (general formulae **9a**–**9c**), dietary supplements, could be absorbed into the bloodstream, which provided new knowledge about how these plant-based saponins are metabolized in vivo [52]. These findings also revealed that both intestinal bacterial metabolism and hydrolysis of ester linkage at C(17)-COOH by carboxylesterases in the liver are the possible deglycosylation metabolism pathways of platycosides in vivo, and indicated their rapid excretion from the organism. The findings are generally important for understanding the metabolism of platycosides after dietary consumption.

Glycyrrhizic acid (**1d**; Figure 1) and glycyrrhetinic acid 3β-d-glucuronide (**10**; Figure 10) have been known to be valuable constituents of licorice. Most recently, they were also isolated from *Glycyrrhiza uralensis* Fisch. (Fabaceae; Chinese licorice), and used in the investigation of uridine diphosphate glucose dehydrogenase activity [53]. The aglycone of both natural products (**1d** and **10**) belongs to the oleanane-type of triterpenoids under the name of glycyrrhetinic acid (**1e**; Figure 1). The plant *G. uralensis* has widely been used in traditional Chinese medicine for treating health disorders, e.g., diabetes [54], hepatitis [55,56,57], bronchitis [58], or AIDS [59]. It also showed a potential therapeutic effect on SARS-CoV-2 [60]. The plant and its extracts of different types have been traditionally used as an anti-inflammatory [61], for liver protection [62] and immune regulation [63], as antivirus [64] and anticancer [65] agents, as well as for immunity regulation. Triterpenoid saponins from *G. uralensis* are predominantly present as glucuronides [66]. In the reviewed original paper [53], uridine diphosphate glucose dehydrogenase (UGDH) activity was studied. The results of molecular docking revealed that five uridine diphosphate glucose (UDPG) isoforms had strong and similar bindings with the substrate. This was the common structural basis enabling the five UDP isoforms to catalyze the substrate. However, there were differences in their binding modes, which indicated different catalytic efficiencies of the UDPG isoforms. All isoforms have strictly conserved residues in the active sites (Ala160 and Glu161), whereas Glu157, Thr163, Asp167, Ser271, and Phe334 were found to be unique amino acid residues of UGDH1–UGDH5, respectively. The binding modes to UGDHs of the two derivatives were similar with UGDH, and they all had multiple identical binding sites. To determine the stability of the whole structure of each of the five UGDH models complexed to the UDPG isoforms, the systems were calculated in silico. The results revealed that the binding forces between the product and UGDHs were relatively weak, and when the enzyme converts the substrate into a product, the product was easier to dissociate from the protein complex [53].

Based on natural saponins **11a** and **11b** (Figure 11) that have been isolated from *Quillaja saponaria* Molina (Quillajaceae; soap bark tree), an evergreen tree that is native to Chile [67], a series of novel oleanane-based saponins (**11c**–**11f**; Figure 11) have been designed using both quillaic acid (**1f**; Figure 1) and echinocystic acid (**1g**; Figure 1) as triterpenoid core molecules. The natural mixture of **11a** and **11b** (Figure 11) has also been known under the code QS-21 [67]. The compounds **11c**–**11f** (Figure 11) were subsequently prepared and immunologically evaluated on the basis of streamlined saponin adjuvants and the Tn (Thomsen-nouveau) carbohydrate antigen [68]. These synthetic compounds (**11c**–**11f**) induced moderate antibody responses in mice, which initiated a search for optimization in the development of self-adjuvanting glycoconjugate cancer vaccines [67,68].

The Tn (Thomsen-nouveau) antigen [*N*-acetylgalactosamine (GalNAc) α-*O*-linked to serine or threonine] is a typical tumor-associated carbohydrate antigen (TACA) overexpressed on the cell surface glycoproteins in human tumors [67,68]. Despite its potential for anticancer vaccine development, it is only weakly immunogenic, and needs to be administered as a conjugate to induce strong immune responses [68]. The classical approach always involved a covalent bond to an immunogenic carrier protein, finally leading to an enhanced antigen presentation, T cell activation, and co-administration with an immunological adjuvant such as the saponin natural products **11a** and **11b**, which were responsible for activating both antibody and cellular immunity, further strengthening the immune response [67]. Compounds **11a** and **11b** represent a mixture of natural triterpenoid glycoside isomers that share a central quillaic acid (**1f**; Figure 1) unit conjugated to a left-hand branched trisaccharide and a C(28)-linked linear tetrasaccharide decorated with a glycosylated acyl chain. This natural mixed product (**11a** and **11b**) has already been widely applied in clinical applications, and has recently got approval as a part of the adjuvant system for vaccines [67]. However, due to the scarcity, heterogeneity, chemical instability, and dose-limiting toxicity of the natural product (a mixture of **11a** and **11b**), its advancement to become a stand-alone adjuvant in vaccines was impeded. This was the main reason why subsequent intensive investigation resulted in designing and developing fully synthetic adjuvant–antigen glycoconjugates **11c**–**11f** as the first example of a di-component vaccine that involved saponins chemically linked to TACAs [67]. The immunological evaluation in mice revealed that this novel saponin–Tn conjugate design induced a certain degree of Tn-specific antibodies in the absence of any external adjuvants or carrier systems. This novel approach may be considered ‘‘self-adjuvanting’’, and it is expected to benefit from additional structural and/or formulation optimization [67].

A modification of the 3β-d-glucuronic acid residue from the *Quillaja* saponin adjuvants was made by the *N*-acylation of the carboxyl group by linear alkyl C_8_-, C_10_-, C_12_-, and C_14_-carbon-chained amines [67,68]. The synthesized amide derivatives bear linear alkyl amine chains as substituents of the carboxyl group of the 3β-d-glucuronic acid unit. This structural modification resulted in a finding that hydrophobic alkyl chains modified the conformation of these glycosides and, subsequently, modified the micellar structures. This finding clearly shows the effect of nano-assembly on pharmacological activity. Structural modifications affected the pharmacological function of the studied compounds in their interactions with cellular receptors, and finally affected the adjuvanticity of the glycosidic compounds. The amide derivatives bearing C_8_ to C_12_ residues modified the response to a pro-inflammatory Th1 immunity. In mice, IgG2a levels were dependent on the direct influence of the secreted interferon-c (IFN-c), a crucial Th1 cytokine. The subsequent derivation by a longer and more lipophilic tetradecylamide group yielded derivatives capable of inducing Th2 immunity, which was demonstrated by a low IgG2a/IgG1 ratio. The immunomodulatory properties or adjuvanticity of the *N*-acylated natural products were affected by the changes in the conformation and micellar structure of the modified molecules. Physicochemical modifications in the structures of the molecules subsequently modified the availability of certain groups, e.g., fucopyranose, to bind to the presumed dendritic cells’ lectin receptor DC-SIGN, an essential step in the stimulation of Th2 immunity. The structural characteristics in an aqueous environment depended on the balance of the hydrophilic and lipophilic moieties (HLB) of glycosides and on the interactions of the newly introduced alkyl chains with the lipophilic triterpenoid aglycone of the native saponin and hydrophilic oligosaccharide chains. All these factors contributed to the explanation of the qualitative and quantitative changes in adjuvanticity of the structurally modified compounds (**11a** and **11b** vs. **11c**–**11f**). The achieved results clearly demonstrated that the correlation of HLB values with the adjuvanticity of the saponins is not straightforward, as it had been proposed earlier [69]. It was found that the participation of different functional groups, such as the triterpenoid aglycone, glycoside residues, and acyl groups, was important both for adjuvanticity and the HLB [69]. A modification of the natural products **11a** and **11b** by long alkyl chains, besides changing their HLB values, yielded novel analogs (**11c**–**11f**) with quantitative and qualitative differences in adjuvanticity [67].

*Aster tataricus* L. f. (Asteraceae) is a nutrient-potent herb found in mainland China, South Korea, and Japan. The rhizomes and roots of *A. tataricus* have been used in traditional medicine to eliminate phlegm and coughing for a long time. The investigation of the plant extracts revealed a potential therapeutic effect of *A. tataricus* as an antioxidant, antitussive, antibacterial, antidepressant, and anti-inflammatory agent, namely due to the abundance of chemical constituents such as shionone, caffeoylquinic acids, and triterpenoid saponins present in the root of the plant [70]. Subsequently, several triterpenoid saponins were isolated from *A. tataricus*, their structures were elucidated, and their potential anti-inflammatory activities were investigated and evaluated by measuring lipopolysaccharide (LPS)-enhanced nitric oxide (NO) formation in murine macrophages [70]. Among these natural products, saponins **12a** and **12b** (Figure 12) exhibited the most potent anti-inflammatory activity (IC_50_ = 42.1 µM and IC_50_ = 1.2 µM, respectively), while the other natural products isolated therefrom were much less active and mostly inactive [70]. Because the quantity of **12b** in the natural source was very low, its synthesis, based on the structure of **12a**, was designed and performed [70,71]. The enhancing anti-inflammatory activity of **12b** (IC_50_ = 1.2 µM) was exceptional within this series of compounds, because other saponins structurally related to **12b** showed only negligible inhibitory activity [70,71]. Inducible nitric oxide synthase (iNOS) and cyclooxygenase-2 (COX-2) protein levels were dose-dependently suppressed by **12b** in the lipopolysaccharide (LPS)-activated RAW 264.7 (murine macrophage) cells. An investigation of the anti-inflammatory mechanism indicated that **12b** reduced the phosphorylation and degradation of the inhibitor of NF-κB, which led to the blocking of NF-κB p65 translocation to the nucleus. This triterpenoid saponin **12b** was worthy of further investigation. However, due to the limited quantity of **12b** that could be isolated from the plant material, its total synthesis was designed [71].

The synthesis of **12b** was achieved by following a [3 + 2] block synthesis strategy, in which the trisaccharide acceptor and the disaccharide donor were rationally designed and obtained from semi-protected monosaccharides by stereoselective glycosylation reactions, either by activation of the thioglycoside or by the formation of glycosyl trichloroacetimidate. The target compound **12b** was then available in a bigger quantity for the subsequent investigation [71]. The bioactivity of saponins depends on their glycoconjugation pattern and, therefore, the carbohydrate being a part of a saponin is always important for the elucidation of the biological activity of saponins, as documented by the enhancing anti-inflammatory activity of **12b** in comparison with that of **12a** [71].

## 3. Triterpenoids with the Ursane Skeleton

In addition to the oleanane-type of triterpenoid saponins mentioned above in the section on triterpenoids **2a**–**2e** with the oleanane skeleton (Figure 2), five ursane-type triterpenoid saponins (**13a**–**13e**; Figure 13) were also isolated from the leaves of *Aralia dasyphylla* Miq. (Araliaceae) [35]. All isolated ursane-type compounds (**13a**–**13e**) were evaluated in the same way as the earlier mentioned oleanane-type plant products **2a**–**2e**. The cytotoxicity of **13a**–**13e** in three human cancer cell lines, human hepatocellular carcinoma (HepG2), human lung adenocarcinoma (LU-1), and human rhabdomyosarcoma (RD) was evaluated in vitro. The molecular docking studies on human glucose transporter 1 (hGLUT1) protein were made in silico. The triterpenoids **13b** and **13d** exhibited good growth inhibition of HepG2 and LU-1 cancer cell lines with IC_50_ values in the range of concentration *c* = 1.76–7.21 µM (Table 2). The in silico molecular docking study results showed that compound **13d** had one of the highest binding affinities to hGLUT1 among these ursane-type plant products. The compounds **13a**–**13e** were also evaluated for their in silico ADMET of absorption, distribution, metabolism, excretion, and oral toxicity parameters. Based on the calculated data, intestinal absorption increased in the series **13e** < **13d** < **13b** < **13c**.

The recent COVID-19 pandemic has been a global threat to public health with emerging attention paid to the SARS-CoV-2 variants, and represented a great challenge to the development of both antiviral agents and vaccines. The ursolic acid-based saponins (**14a** and **14b**; Figure 14) were subjected to an investigation for novel SARS-CoV-2 fusion inhibitors [72]. Structurally, these saponins bear a hydrophilic branched trisaccharide α-l-rhamnopyranosyl-(1→2)-[α-l-rhamnopyranosyl-(1→4)]-β-d-glucopyranosyl residue, known as chacotriose, incorporated to the hydrophobic aglycone (**1b**) by the β-d-glycosidic linkage, followed by different side chains at the C(17)-COOH group of the ursolic acid. The β-chacotriosyl moiety of **14a** and **14b** was essential for biological activity. The compounds **14a** and **14b** showed inhibition rates higher than 80% in the antiviral assay at a concentration of *c* = 40 µM. In turn, modifying the β-chacotriosyl moiety in the α-l-rhamnopyranosyl-(1→2)-β-d-glucopyranosyl or the α-l-rhamnopyranosyl-(1→4)-β-d-glucopyranosyl residues resulted in a significant loss of inhibition effect. This result illustrated the crucial role of the β-chacotriosyl moiety in the anti-SARS-CoV-2 activity. However, neither chacotriose nor ursolic acid (**1b**) showed inhibition toward SARS-CoV-2 as single compounds at a concentration of *c* = 40 µM. This finding demonstrated that these saponins acted as integral structures, but neither the oligosaccharide chain nor the aglycone alone was capable of generating anti-SARS-CoV-2 activity.

To investigate the effect of different substituents in different locations of the basic structure of the triterpenoid molecule, a series of the title saponins **14c**–**14q** (Figure 14) were designed and synthesized. Based on the antiviral effect of **14a** (EC_50_ = 10.69 µM), the inhibition rates of the title saponins against SARS-CoV-2 at the concentrations of *c* = 40 µM (high concentration) and *c* = 10 µM (low concentration; EC_50_ values) were evaluated. The incorporated trifluoromethoxy group as a bioisostere of the methoxy group might form an additional potential interaction with the S protein. The trifluoromethoxy group was first incorporated at the *ortho*-position of the phenyl ring to generate **14c**; however, the novel compound displayed higher cytotoxicity than **14a** (Table 3). The impact of 2,6-disubstitution was determined with **14d**. This structural modification resulted in a slight improvement in biological activity, comparable with that of **14a**, and the increased antiviral activity was not accompanied by cytotoxicity, as followed from the CC_50_ value of the concentration of *c* > 100 µM against HEK293T human-ACE2 cells with **14d**. In contrast, a replacement of the phenyl ring of **14a** with a 2,3-dihydrobenzo[*b*][1,4]dioxine residue (**14e**) had a negative effect on the antiviral activity. A replacement of the phenyl group by more sterically hindered biphenyl moieties (**14f**–**14h**) resulted in a loss of biological activity. The compound **14i**, bearing a quinolone ring with a similar steric hindrance of the substituent, showed surprisingly analogous inhibition activity as **14a**. A similar increase in inhibitory potency was also observed with other (6 + 5)-fused heteroaryl derivatives containing the nitrogen group (**14k** was more potent than **14a** in cellular assays). These results revealed that larger substituents at the C(28)-position of the ursolic acid (**1b**; Figure 1) were unfavorable substituents, and finally directed the investigation towards smaller substituents. However, attempts to lower steric hindrance by substituting the phenyl side chain with a small tetrazole group (**14l**) induced strong toxicity towards HEK293T human-ACE2 cells. The length of the alkyl spacer between the amide group and the phenyl ring affected the inhibition effects of the compounds as well (**14m**–**14o**) [72].

In summary, the leading compound **14d** showed potent antiviral activity against infectious SARS-CoV-2 (Wuhan-HU-1 variant) in Vero-E6 cells and was also effective against the infection of diverse pseudo-typed SARS-CoV-2 variants with mutations in the S protein, including the Omicron and Delta variants. Compound **14d** was also able to target the cavity between S1 and S2 subunits to stabilize the pre-fusion state of the SARS-CoV-2 S protein that resulted in interfering with virus–cell membrane fusion. This investigation introduced a series of novel SARS-CoV-2 fusion inhibitors (**14c**–**14q**) against SARS-CoV-2 and its variants based on the (3β)-3-*O*-chacotriosyl derivatives of the ursolic acid (**1b**; Figure 1) skeleton. On the basis of **14a**, subsequent chemical optimization led to the development of the novel and potent lead compound **14d**, which had an excellent potency (EC_50_ = 2.05 µM) and a favorable SI value (SI > 49) when tested with infectious SARS-CoV-2, and displayed a broad-spectrum entry inhibition against recently emerged SARS-CoV-2 variants, such as Delta and Omicron. It has been found that, by utilizing surface plasmon resonance (SPR) measurement, the co-immunoprecipitation (Co-IP) assay, cell–cell fusion assay, and docking studies in combination with mutagenesis studies, the most promising was the compound **14d** from this series. The compound **14d** showed the capability of occupying the cavity between S1 and S2 subunits in the SARS-CoV-2 S protein for interfering with virus–cell fusion, finally resulting in a broad and effective antiviral activity in vitro. The results supported further clinical development of the fusion inhibitors structurally related to **14d** with a high inhibition effect on SARS-CoV-2 and its variants [72].

*Ilex pubescens* Hook. & Arn. (Aquifoliaceae; holly) represents another medicinal plant that is a source of triterpenoid saponins with the ursane-type of skeleton (**15a**–**15d**; Figure 15) [73]. The extract from the plant tissues is capable of regulating lipid levels, such as lysophosphatidylcholine (LPC) and lysophosphatidylethanolamine (LPE). The studied triterpenoid saponins isolated from *I. pubescens* improved blood biochemical function in the process of blood stasis syndrome, and played a role in vascular protection and maintenance of the normal morphology of blood vessels [73]. This study indicated that combining systemic pharmacology, metabolomics, and molecular docking is an effective and feasible strategy to discover potential therapeutic targets of herbal medicines [73]. During this investigation, it was found that the ursane-type of triterpenoid saponins **15a**–**15d** from *I. pubescens* generally improved blood biochemical function in the process of blood stasis syndrome, and played a role in vascular protection and maintenance of the normal morphology of blood vessels. Metabolite pathways involved in steroid biosynthesis and sphingolipid metabolism were significantly disturbed. Both metabolomics analysis and network pharmacology results showed that triterpenoid saponins from *I. pubescens* ameliorated vascular injury. The lipid accumulation was mediated by the PI3K/AKT signaling pathway activation. The simulation of molecular dynamics and the enzyme inhibitory activity resulted in a finding that the main components of triterpenoid saponins from *I. pubescens* (**15a**–**15d**) gave rise to stable complexes with PI3K, AKT, and eNOS that displayed significant binding affinity. It was noted that levels of PI3K, AKT, p-AKT, eNOS mRNA, and other proteins were always considerably elevated when treated with triterpenoid saponins from *I. pubescens* (**15a**–**15d**). Therefore, these triterpenoid saponins protected the vasculature by regulating the PI3K/AKT signaling pathway, activating eNOS and increasing the release of NO [73].

## 4. Triterpenoids with the Lupane Skeleton

Acankoreagenin, (3α)-3-hydroxy-lup-20(29)-en-23,28-dioic acid (**16a**), and impressic acid, (3α,11α)-3,11-dihydroxylup-20(29)-en-28-oic acid (**16b**), are two lupane-type triterpenoids that were isolated from various *Acanthopanax* (*Eleutherococcus*) Decne. & Planch. (Araliaceae; thorny ginseng) and *Schefflera* J.R. Forst. & G. Forst. (Araliaceae; umbrella tree) species (Figure 16) [74]. The antinociceptive and anti-inflammatory activities of *Schefflera octophylla* extracts are well known and have been characterized. Acankoreagenin (**16a**) demonstrated inhibitory activities towards different enzymes responsible for diabetes, such as α-glucosidase and protein tyrosine phosphatase 1B (PTP1B), displaying IC_50_ ∼ 13 and 16 µM, respectively, and towards α-amylase (in vitro; IC_50_ = 31 µM). This compound reduced the production of the cytokine-stimulated inducible nitric oxide synthase (iNOS) and caused a limited activation of the transcription factor NF-κB in cells. The inhibition of the NF-κB pathway prevents iNOS expression in vitro [74]. The compound **16a** displayed anti-inflammatory activity both in vitro and in vivo, reducing the serum levels of inflammatory cytokines (TNF-α and IL-1β) and the release of the protein HMGB1, a pro-inflammatory cytokine, in a dose-dependent way [74].

More than 15 acankoreosides (**16c**–**16q**; Figure 16) were derived from the acankoreagenin aglycone. Compounds such as **16c** and **16d** were proven to act as remarkable anti-inflammatory agents, inhibiting cytokine release from the activated macrophages. Regardless of their effectiveness, acankoreosides and impressic acid (**16b**) have so far been much less intensively studied than the structurally related compounds betulinic acid (**1c**) and 23-hydroxybetulinic acid (anemosapogenin). The structural differences (notably the *R*/*S* stereoisomerism of the C(3)-OH group of the triterpenoid skeleton) and functional similarities of these compounds were investigated. The complete series of acankoreosides was presented for the first time in the paper by Bailly [74]. These natural products were also investigated as anti-inflammatory agents, and acankoreosides have been recommended as templates for designing new anticancer and antiviral drugs.

Russian authors [75] synthesized an interesting mannopyranosyl derivative of betulinic acid, decorated by a phosphoniohexyl group that formed an ester at the C(17)-COOH group. The conjugation of betulinic acid (**1c**) or its saponine derivative with vector fragments, lipophilic delocalized cations, e.g., triphenylphosphonium salt, can provide targeted delivery of the drug agent to the required organs or tissues, and/or selective interaction with a certain type of transformed cells. This approach achieved an increase in the concentration of the active agent in the mitochondrial matrix by more than 1000 times [76]. The inclusion of a phosphonium fragment into the structure of betulinic acid-based mannopyranoside saponin (**17**; Figure 17) led to a significant increase in the cytotoxicity of the target compound **17** in human cancer cell lines, in comparison with the parent betulinic acid (**1c**; Figure 1) [75,76]. Moreover, it was shown in Canada that decorating the C(3)-OH group of betulinic acid (**1c**) with the d-mannopyranoside or l-rhamnopyranoside units led to a substantial enhancement of cytotoxicity in the respective triterpenoid saponins in comparison with the parent betulinic acid (**1c**) [77]. Finally, the Russian authors combined both structural motifs into a single structure **17**, in which they used the α-linked d-mannopyranoside unit to produce saponin, and synthesized a highly cytotoxic compound **17** [75].

## 5. Conclusions

We have reviewed the most recent triterpenoid (oleanane-, ursane-, and lupane-type) saponins found in plant sources, which have been identified and investigated. Table 4 summarizes all relevant data, including references. The types of pharmacological effects found during the investigation of the plant products represent a broad spectrum of biological activity. The mode of action of the compounds was mentioned wherever it was described in the original literature source. In addition to naturally-occurring plant products, synthetic glycoconjugates related to natural saponin products were mentioned and evaluated. The semisynthetic derivation of natural triterpenoids and triterpenoid saponins resulted in compounds displaying enhanced pharmacological effects, e.g., **11c**–**11f** vs. **11a** and **11b** or **12b** vs. **12a**. Natural products have been a source of inspiration for the synthesis of novel compounds showing novel or enhanced types of pharmacological effects (**14c**–**14q** or **17**). Adjuvant–antigen glycoconjugates **11c**–**11f** represent the first example of a di-component vaccine that involved saponins chemically linked to the tumor-associated carbohydrate antigen overexpressed on the cell surface glycoproteins in human tumors, and those compounds already show a good prerequisite for obtaining approval for practical application in the form of anticancer vaccines [30].

Based on the number of original and review papers targeting plant triterpenoids that have been published in the most recent 5 years, a clear conclusion can be postulated on the increasing importance of sustainable resources of future therapeutics. A broad scale of human diseases that may be successfully treated by the semisynthetic derivatives of triterpenoids has been ever increasing. Several of the most successful triterpenoid derivatives have already been in practical use. Among them, the betulinic acid-based anti-HIV agent, bevirimat, should be highlighted [78,79]. It is expected that the number of practically successful triterpenoid derivatives will gradually increase. Our recent results dealing with different derivatives of plant triterpenoids, not including their glycoconjugates, displayed the high potential of these compounds as antimicrobial, antiviral, and cytotoxic agents [26,27,28]. Some of them showed their capability of acting as chemodynamic and photodynamic therapy agents [80,81] or coordinating metal ions, such as radioisotopic ^64^Cu(II) salts, for potential positron emission tomography (PET) imaging and radiotherapy in combination with cytotoxicity [27]. Due to the important potential of plant triterpenoids and their semisynthetic derivatives, they have been intensively studied as functional nanoscale assemblies with an impact in the areas of nanomedicine and drug delivery, but also in dye removal or catalysis in different chemical processes, and this importance will be even increasing in the forthcoming time [28].

## Figures and Tables

**Figure 1 pharmaceuticals-16-00386-f001:**
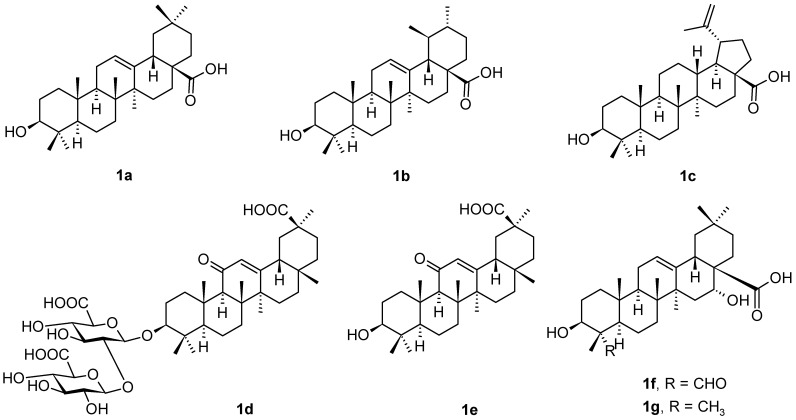
The structures of selected natural triterpenoids.

**Figure 2 pharmaceuticals-16-00386-f002:**
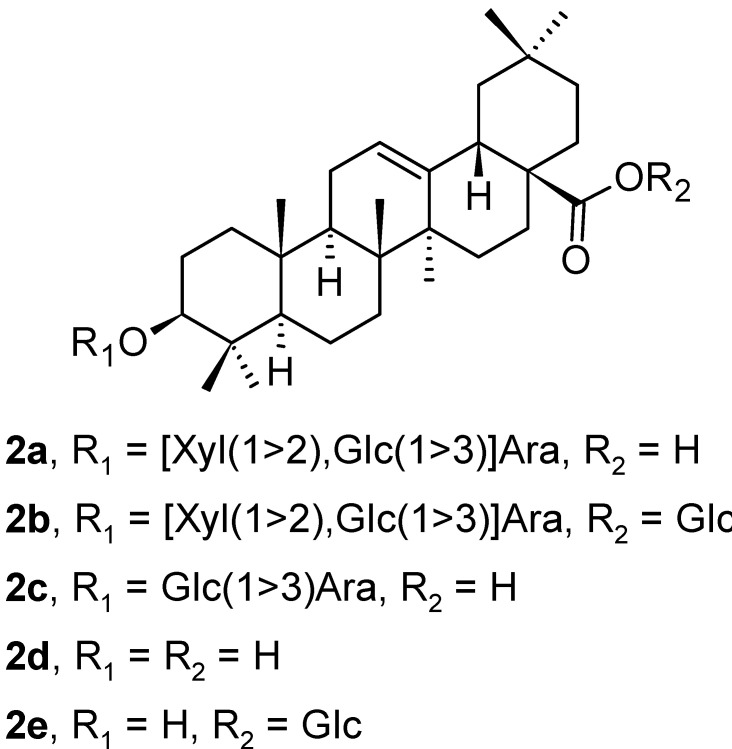
Oleanane-type triterpenoids found in the leaves of *Aralia dasyphylla*.

**Figure 3 pharmaceuticals-16-00386-f003:**
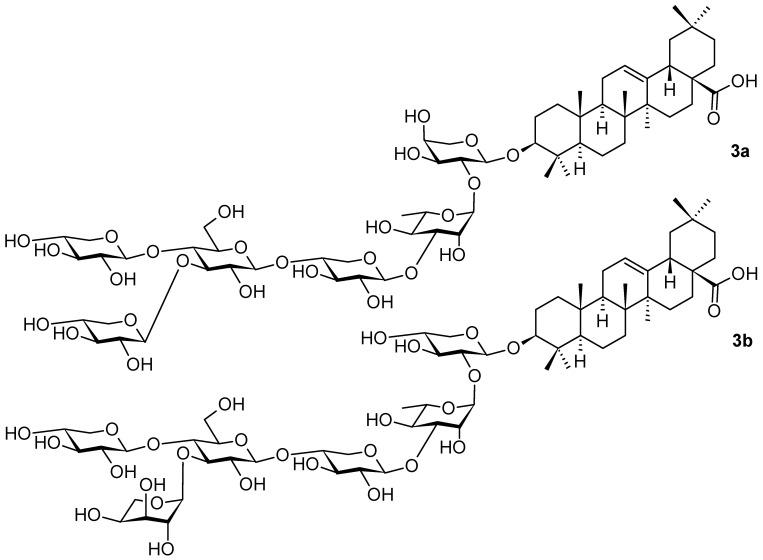
Oleanane-type triterpenoids found in the plants of the genus *Weigela* (part 1).

**Figure 4 pharmaceuticals-16-00386-f004:**
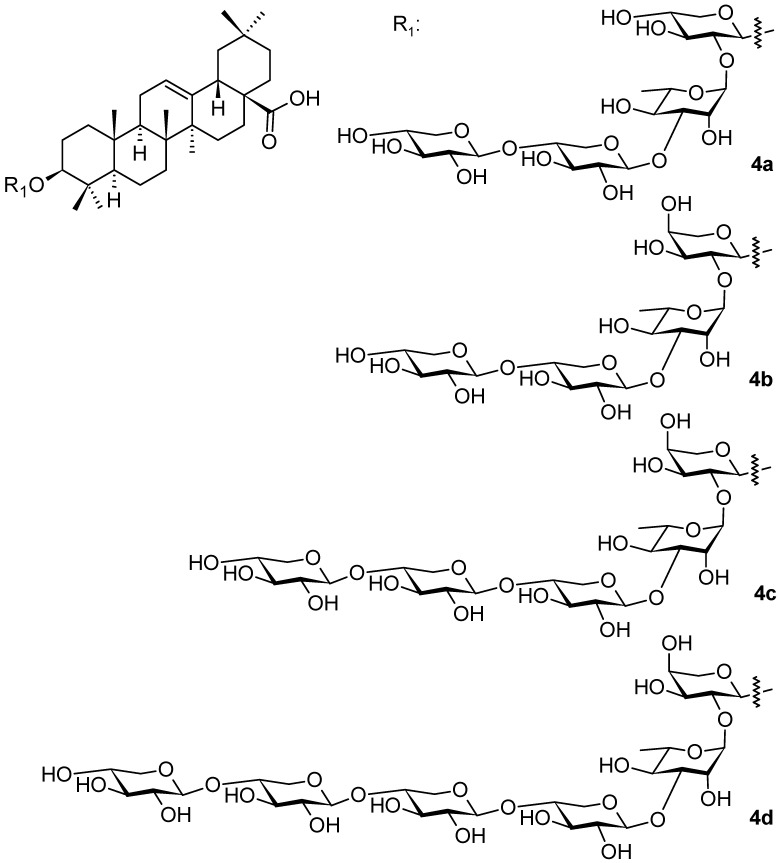
Oleanane-type triterpenoids found in the plants of the genus *Weigela* (part 2).

**Figure 5 pharmaceuticals-16-00386-f005:**
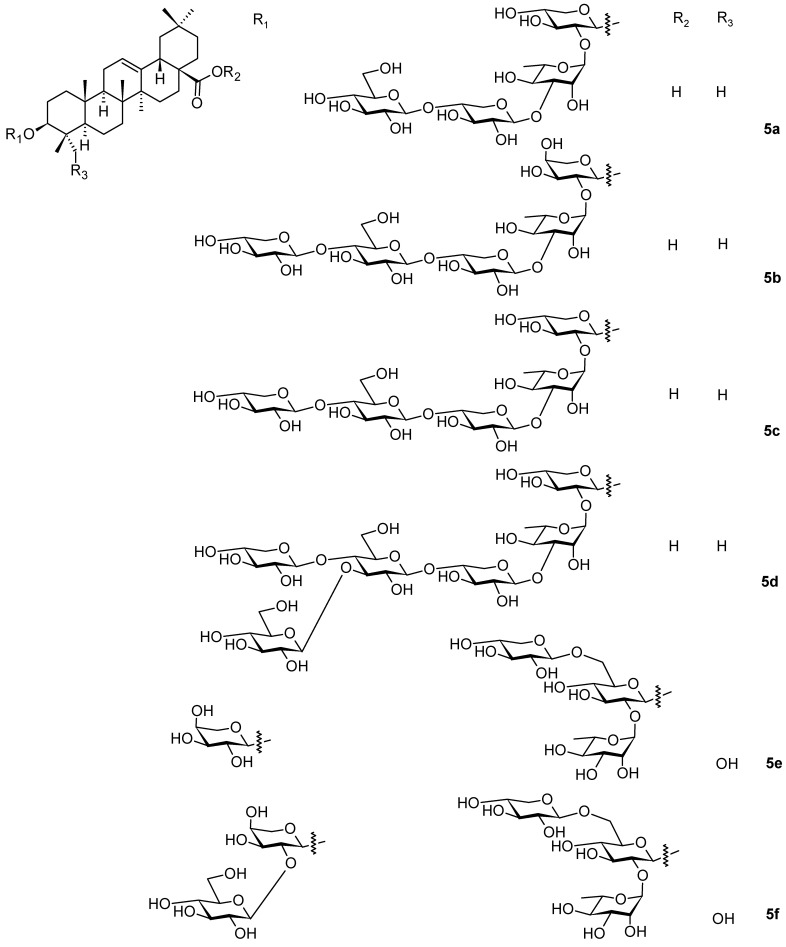
Oleanane-type triterpenoids found in the plants of the genus *Weigela* (part 3).

**Figure 6 pharmaceuticals-16-00386-f006:**
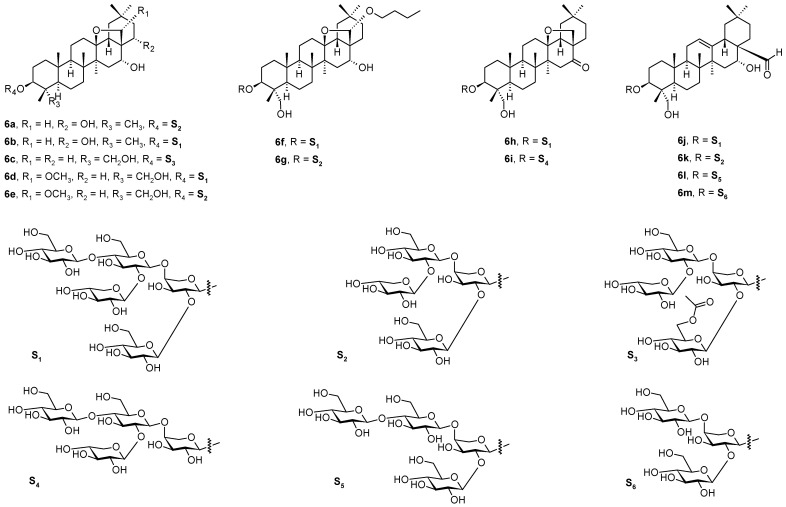
Monellosides **6a**–**6m** from *Anagallis monelli* ssp. *linifolia*.

**Figure 7 pharmaceuticals-16-00386-f007:**
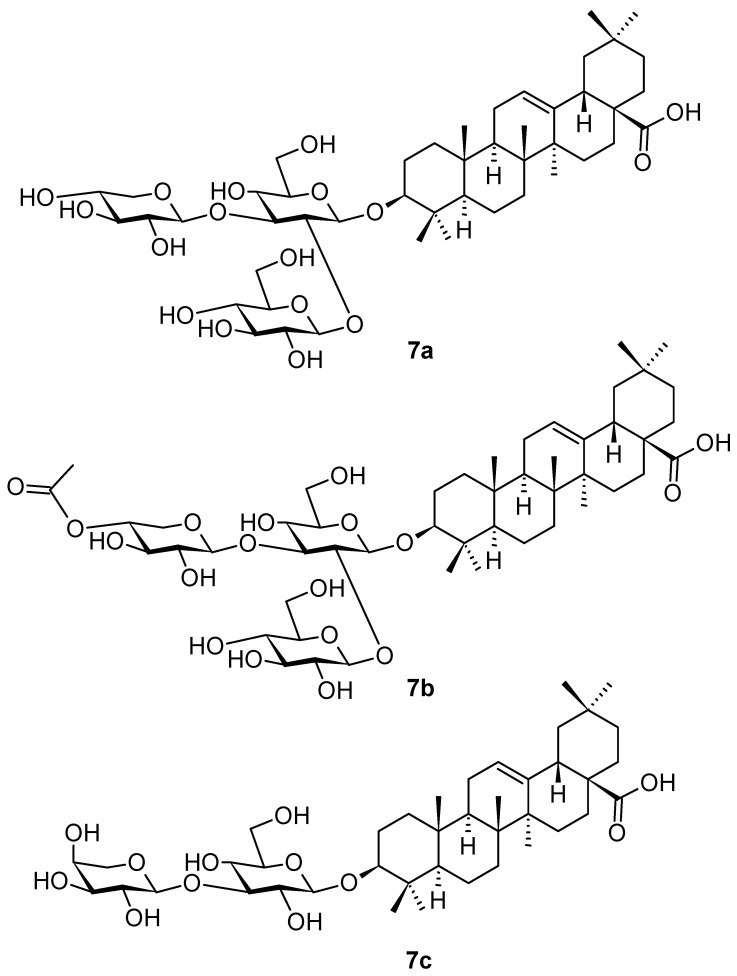
Oleanane-type lepiginosides **7a**–**7c** from the stem bark of *Lepisanthes rubiginosa*.

**Figure 8 pharmaceuticals-16-00386-f008:**
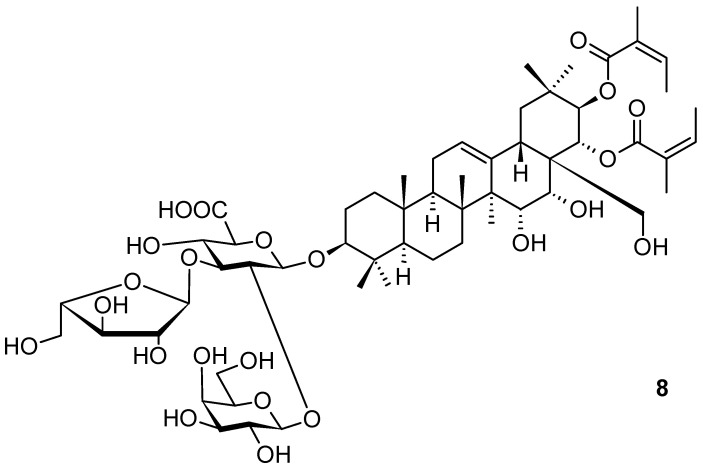
Oleanane-type xanthoceraside **8** from *Xanthoceras sorbifolium*.

**Figure 9 pharmaceuticals-16-00386-f009:**
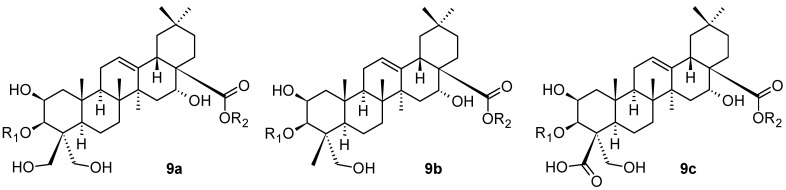
Oleanane-type platycosides from *Platycodon grandiflorus*.

**Figure 10 pharmaceuticals-16-00386-f010:**
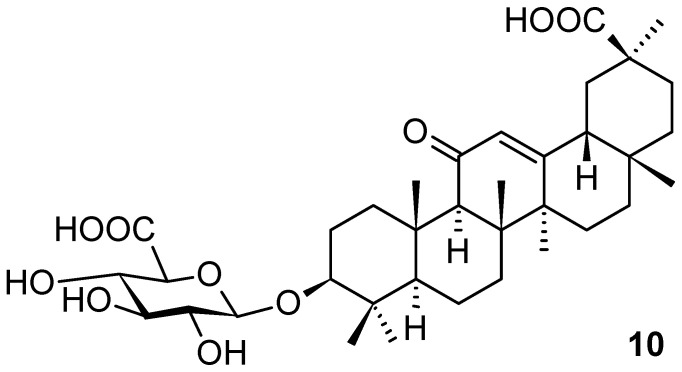
Glycyrrhetinic acid 3β-d-glucuronide.

**Figure 11 pharmaceuticals-16-00386-f011:**
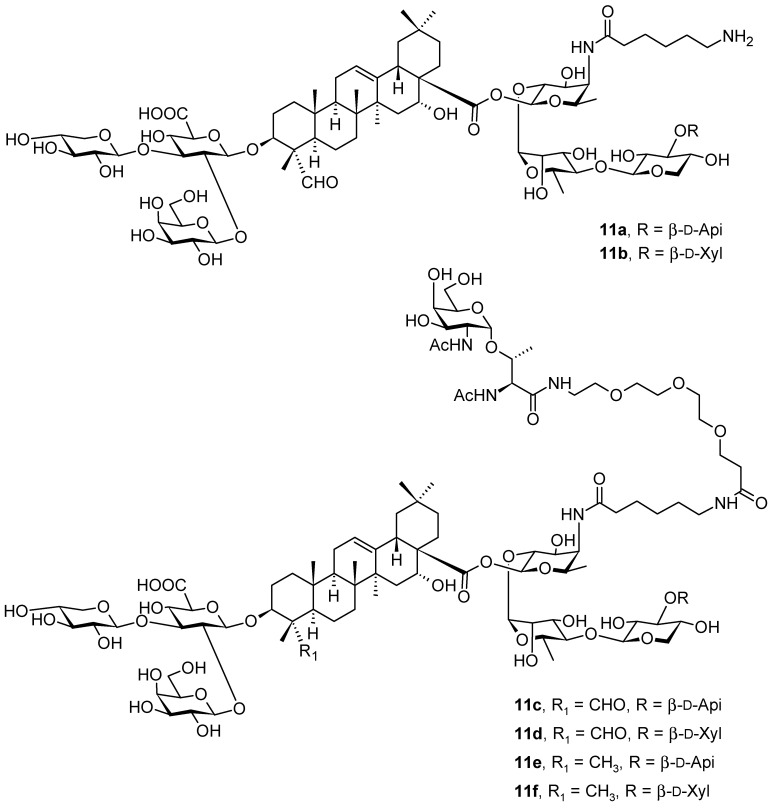
Oleanane-type saponin adjuvants.

**Figure 12 pharmaceuticals-16-00386-f012:**
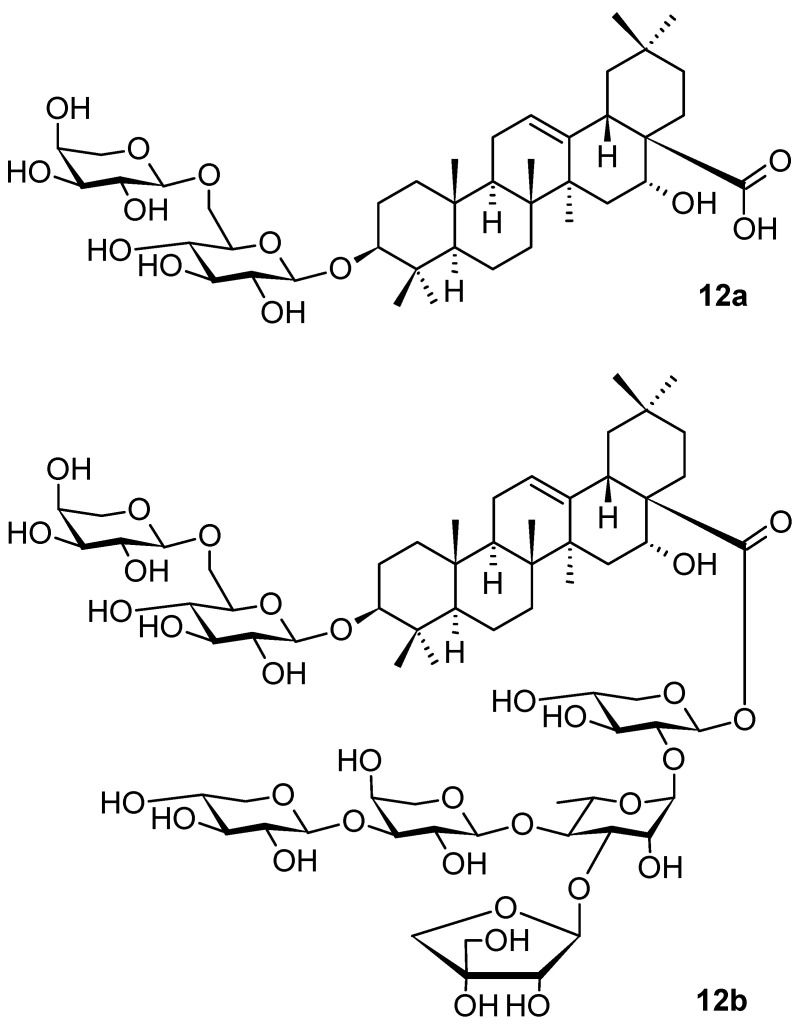
Oleanane-type triterpenoids from *Aster tataricus*.

**Figure 13 pharmaceuticals-16-00386-f013:**
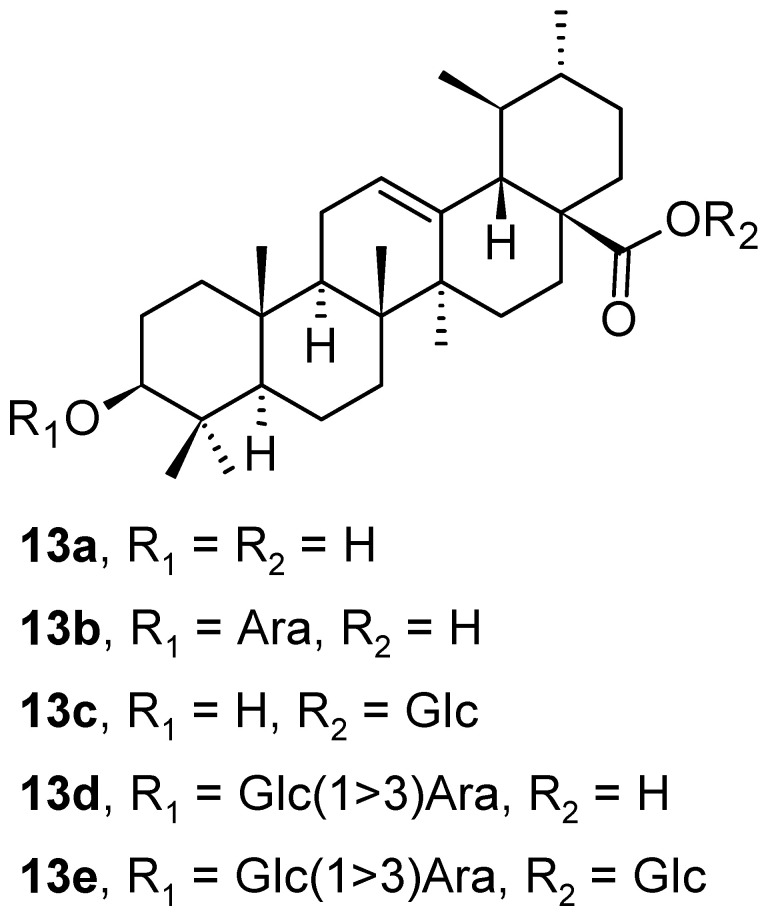
Ursane-type triterpenoids found in the leaves of *Aralia dasyphylla*.

**Figure 14 pharmaceuticals-16-00386-f014:**
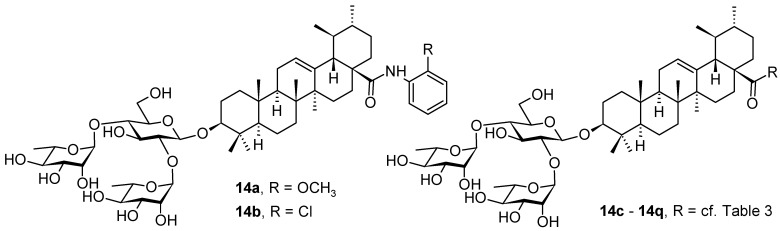
Ursane-type saponin derivatives for the SARS-CoV-2 treatment.

**Figure 15 pharmaceuticals-16-00386-f015:**
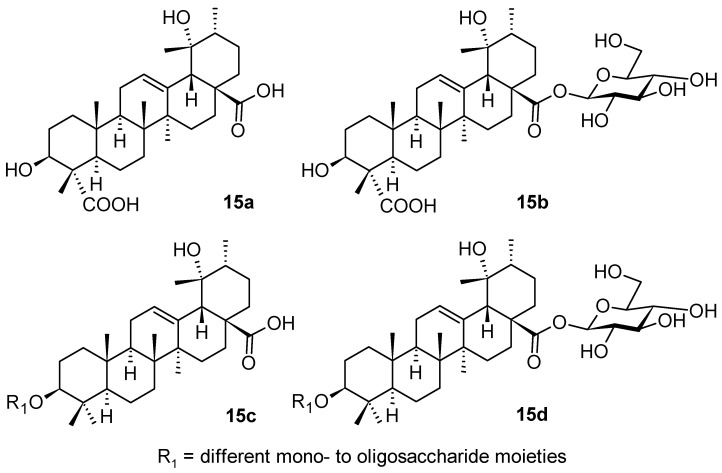
Ursane-type triterpenoid saponins from *Ilex pubescens*.

**Figure 16 pharmaceuticals-16-00386-f016:**
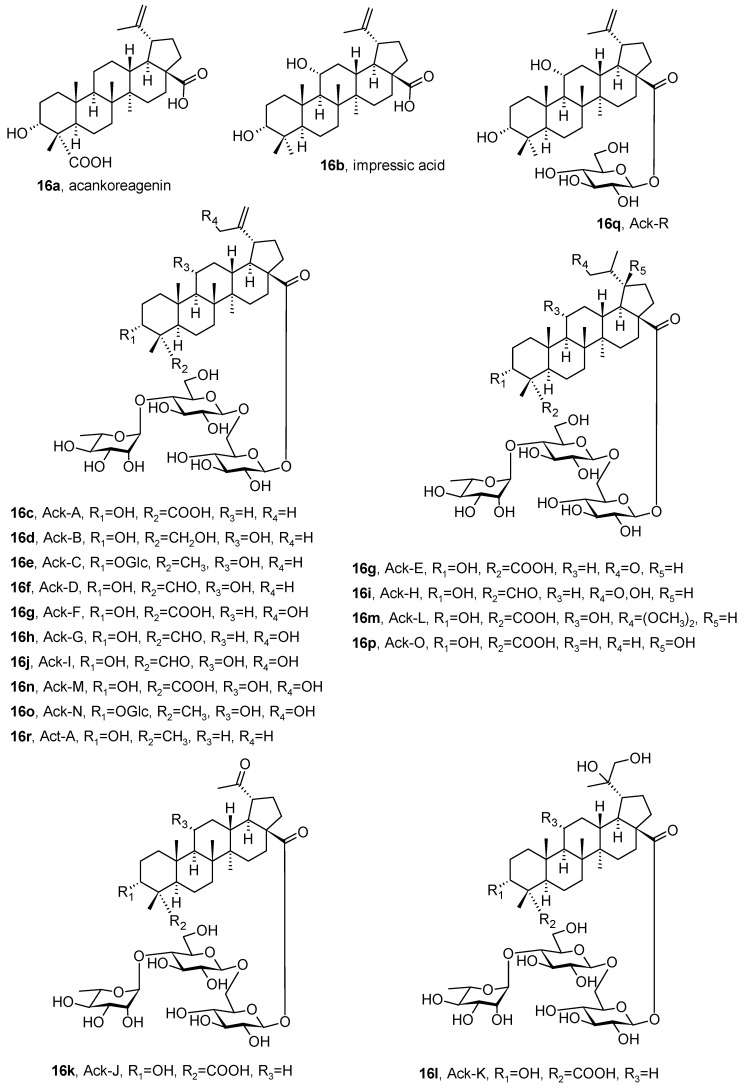
Lupane-type acankoreagenin, impressic acid, and acankoreosides from various *Acanthopanax* (*Eleutherococcus*) species.

**Figure 17 pharmaceuticals-16-00386-f017:**
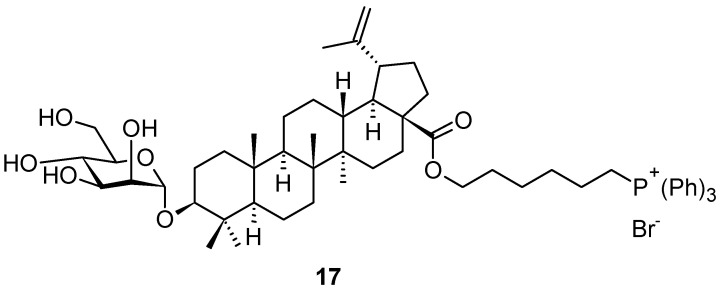
Lupane-type triterpenoid saponin derivative.

**Table 1 pharmaceuticals-16-00386-t001:** The cell survival values and in vitro cytotoxicity of **2a**–**2e** in three different cancer cell lines [35].

Compound	CS Values [%] (Mean ± SD) ^a^			IC_50_ [μM] (Mean ± SD) ^b^		
	HepG2 ^c^	LU-1 ^d^	RD ^e^	HepG2 ^c^	LU-1 ^d^	RD ^e^
**2a**	0	18.51 ± 1.20	67.13 ± 2.17	3.24 ± 0.22	2.55 ± 0.12	>100 ± 1.45
**2b**	97.01 ± 0.90	76.47 ± 2.00	95.57 ± 1.90	>100 ± 1.32	>100 ± 0.97	>100 ± 0.58
**2c**	5.98 ± 0.39	0	0	2.73 ± 0.12	1.76 ± 0.11	2.63 ± 0.10
**2d**	43.98 ± 1.46	25.21 ± 1.34	72.82 ± 1.26	7.21 ± 0.57	4.56 ± 0.24	>100 ± 1.28
**2e**	60.88 ± 1.80	67.52 ± 2.33	65.52 ± 1.54	>100 ± 2.10	>100 ± 1.43	>100 ± 1.45
ellipticine ^f^	1.25 ± 0.30	1.87 ± 0.20	0	1.22 ± 0.09	1.30 ± 0.10	1.18 ± 0.08

^a^ The concentration of the sample *c* = 5 µg · mL^−1^. CS (cell survival) value [%] is the ability of cells to survive at a certain concentration of the reagent [in %] compared with the control (*n* = 3); ^b^ Data are presented as means of the concentration of the sample required for 50% inhibition of cell growth ± SD from triplicated; ^c^ HepG2 (human hepatocellular carcinoma); ^d^ LU-1 (human lung adenocarcinoma); ^e^ RD (human rhabdomyosarcoma); ^f^ ellipticine (5,11-dimethyl-6*H*-pyrido[4,3-*b*]carbazole) was used as a positive control.

**Table 2 pharmaceuticals-16-00386-t002:** The cell survival values and in vitro cytotoxicity of **13a**–**13e** in three different cancer cell lines [35].

Compound	CS Values [%] (Mean ± SD) ^a^			IC_50_ [μM] (Mean ± SD) ^b^		
	HepG2 ^c^	LU-1 ^d^	RD ^e^	HepG2 ^c^	LU-1 ^d^	RD ^e^
**13a**	68.42 ± 0.96	29.61 ± 0.15	66.79 ± 1.51	>100 ± 1.45	7.04 ± 0.64	>100 ± 1.16
**13b**	45.98 ± 1.45	25.11 ± 1.54	72.81 ± 1.56	7.21 ± 0.60	4.56 ± 0.18	>100 ± 0.65
**13c**	59.88 ± 1.80	65.52 ± 2.53	64.52 ± 1.34	>100 ± 1.80	>100 ± 0.84	>100 ± 0.35
**13d**	37.20 ± 2.30	15.12 ± 0.60	70.00 ± 2.19	5.36 ± 0.47	2.85 ± 0.20	>100 ± 1.42
**13e**	98.28 ± 0.95	78.70 ± 1.15	98.42 ± 1.47	>100 ± 1.36	>100 ± 1.34	>100 ± 1.89
ellipticine ^f^	1.25 ± 0.30	1.87 ± 0.20	0	1.22 ± 0.09	1.30 ± 0.10	1.18 ± 0.08

^a^ The concentration of the sample *c* = 5 µg · mL^−1^. CS (cell survival) value [%] is the ability of cells to survive at a certain concentration of the reagent [in %] compared with the control (*n* = 3); ^b^ Data are presented as means of the concentration of the sample required for 50% inhibition of cell growth ± SD from triplicated; ^c^ HepG2 (human hepatocellular carcinoma); ^d^ LU-1 (human lung adenocarcinoma); ^e^ RD (human rhabdomyosarcoma); ^f^ ellipticine (5,11-dimethyl-6*H*-pyrido [4,3-*b*]carbazole) was used as a positive control.

**Table 3 pharmaceuticals-16-00386-t003:** Antiviral activity of the saponins **14c**–**14q** tested in SARS-CoV-2 S protein at the concentrations of *c* = 10 µM and *c* = 40 µM, respectively [72].

Compound	R	Inhibition Rate [%]
		*c* = 10 μM	*c* = 40 μM
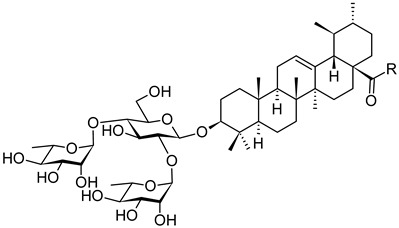
**14c**	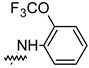	toxic ^a^	toxic ^a^
**14d**	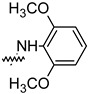	96.84 ± 5.40	46.81 ± 3.55
**14e**	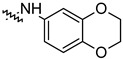	30.45 ± 2.13	−2.49 ± 0.24
**14f**	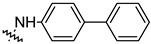	27.44 ± 4.11	16.94 ± 2.35
**14g**	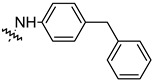	19.25 ± 1.03	10.32 ± 2.10
**14h**	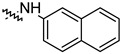	14.51 ± 1.06	−7.27 ± 1.03
**14i**	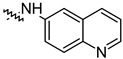	33.08 ± 1.02	27.35 ± 2.51
**14j**	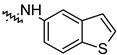	−21.66 ± 1.20	−31.75 ± 4.32
**14k**	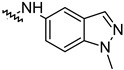	51.87 ± 2.23	41.08 ± 1.89
**14l**	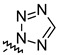	toxic ^a^	toxic ^a^
**14m**	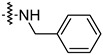	92.49 ± 3.78	59.23 ± 4.21
**14n**	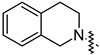	58.74 ± 5.13	32.52 ± 2.59
**14o**	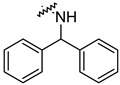	1.58 ± 0.34	0.47 ± 0.08
**14p**	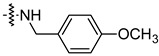	85.69 ± 1.33	36.26 ± 2.45
**14q**	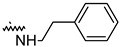	49.47 ± 3.08	28.16 ± 2.22
salvianolic acid C ^b^		98.30 ± 2.62	80.25 ± 2.56

^a^ the title saponins showed significant cytotoxicity at corresponding concentrations, and the inhibition rate could not be calculated; ^b^ salvianolic acid C, (2*R*)-3-(3,4-dihydroxyphenyl)-2-[(*E*)-3-[2-(3,4-dihydroxyphenyl)-7-hydroxy-1-benzofuran-4-yl]prop-2-enoyl]oxypropanoic acid, was used as a reference compound.

**Table 4 pharmaceuticals-16-00386-t004:** Summary of the reviewed triterpenoid saponins.

Plant [Reference]	Saponin Found	Type of Pharmacological Application
*Aralia dasyphylla* Miq. (Araliaceae) [35]	Oleanane-type	Cytotoxicity
*Weigela* Thunb. (Caprifoliaceae) [36,45]	Oleanane-type	Cytotoxicity, antifungal activity, antibacterial activity
*Anagallis monelli* ssp. *linifolia* (L.) Maire (Primulaceae) [46]	Oleanane-type: monellosides	Cytotoxicity, antibacterial, antifungal, antioxidant, antihyperglycemic and antipruritic activity
*Lepisanthes rubiginosa* Roxb. (Sapindaceae) [47]	Oleanane-type	Antibacterial activity
*Xanthoceras sorbifolium* Bunge (Sapindaceae) [50]	Oleanane-type: xanthoceraside	Major depressive disorders treatment
*Platycodon grandiflorus* Jacq. (Campanulaceae) [52]	Oleanane-type: platycosides	Dietary supplements absorbed into the bloodstream
*Glycyrrhiza uralensis* Fisch. (Fabaceae) [53] ^a^	Oleanane-type: glycyrrhizic acid, glycyrrhetinic acid glucuronide	Anti-inflammatory activity, liver protection, immune regulation, antiviral activity, anticancer activity
*Quillaja saponaria* Molina (Quillajaceae) [67,68]	Oleanane-type	Saponin adjuvants, carbohydrate antigen
*Aster tataricus* L. f. (Asteraceae) [70,71]	Oleanane-type	Anti-inflammatory activity
*Aralia dasyphylla* Miq. (Araliaceae) [35]	Ursane-type	Cytotoxicity
*Ilex pubescens* Hook. & Arn. (Aquifoliaceae) [73]	Ursane-type	Regulation of lipid level, improving blood biochemical function
*Acanthopanax* spp. (*Eleutherococcus*) Decne. & Planch. (Araliaceae) [74]	Lupane-type: acankoreagenin, impressic acid, acankoreosides	Antinociceptive activity, anti-inflammatory activity
*Schefflera* spp. J.R. Forst. & G. Forst. (Araliaceae) [74]	Lupane-type: acankoreagenin, impressic acid, acankoreosides	Antinociceptive activity, anti-inflammatory activity

^a^ Discovery [53] mentioned in this review paper, because glycyrrhizic acid and glycyrrhetinic acid β-d-glucuronide are known from many plant sources.

## Data Availability

Not applicable.

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
