# Peer review of "Saponins of Selected Triterpenoids as Potential Therapeutic Agents: A Review"

_pharmaceuticals, 2023, doi:10.3390/ph16030386_

Round 1

Reviewer 1 Report

Authors tried to write a review  which is entitled with an important topic called “natural  Therapeutic Agents. Indeed, the use of Therapeutic Agents in the treatment of various diseases is a research direction of upmost importance. However, the present review has been written very poorly. It provides only superficial discussion in general terms without useful insight. The level and impact of the provided discussion is very low and does not reach the requirements of a good journal. The tendencies mentioned in the text and in the conclusions have already been described in the literature even with better way, and this review does not give a new insight.  It is just poor summary of some published literature. More importantly, the same group has published TWO reviewer papers with very similar topic in 2022 (reference no. 3 and 29). Compared to these two review papers, the current manuscript does not contain any new information. Authors also mentioned it in the abstract.  In Table 1, 2 & 3, authors provided some data. Are those unpublished data from the groups? If published, authors must provide the references.

Author Response

Response to reviewer 1:

Thank you for your valuable comments and suggestions. In a response to the criticism, I would like to stress that neither of the reviews mentioned by you (references No. 3 and 29, now No. 3 and 28) has dealt with saponins. Our review article (ref. No. 3) has dealt with amides and other nitrogen-containing structural modifiers of several triterpenoids, and the other review article of ours (ref. No. 28) has dealt with supramolecular nanostructures formed by different derivatives of triterpenoids than triterpenoid saponins. The similarity mentioned by the reviewer is only based on the fact that natural plant triterpenoids were focused as natural sources of plant products employed in our investigations.

References were added to the heading of the Tables 1 - 3 to avoid misunderstanding. Those references had already been in the text of the originally submitted manuscript, in the parts of the text of the manuscript dealing with the specific compounds that appeared also in the Tables 1-3.

A new paragraph and Table 4 were added to the Conclusion to summarize the reviewed triterpenoid saponins. The most important achievements mentioned in this review paper were evaluated.

Reviewer 2 Report

Title of review: Saponins of Selected Triterpenoids as Potential Therapeutic 2 Agents: A Review.

This review is focussing on saponins of the oleanane, ursane and lupane types of triterpenoids that include several plant triterpenoids displaying various important pharmacological effects.

But their importance, due to the general ability of triterpenoid compounds to self-assemble into nanoscale materials, is not much discussed.

The comments/suggestions are highlighted in Manuscript.

Author Response

Response to reviewer 2:

Thank you for your valuable comments and suggestions. In a response to your criticism on self-assembly, I would like to stress that the literature data on self-assembly of triterpenoid saponins are very rare, and with most triterpenoid saponins this topic has not yet been studied at all.

The Abstract of the manuscript was changed in the criticized part. Self-assembly was deleted from the abstract and from the keywords. In turn, more information on enhancing pharmacological effect was introduced.

The suggested Table of phytoconsituents (Table 4) was produced. It is presented within the Conclusion of the manuscript. A new paragraph was introduced into the Conclusion, summarizing the reviewed plant products and synthesized glycoconjugates.

We apologize for several typing errors you discovered in the original manuscript. They were corrected.

Reviewer 3 Report

The review manuscript presented deals with works on saponines of selected triterpenoids as potential therapeutic agents.

On a first glimpse I was a little bit surprised by the short period the review article covers (works from 2019 - 2022). As presented in the instruction section various review articles exist at least covering parts of the broad field of tritepenoid chemistry and/or application.

But after reading the manuscript I think the content is of sufficient importance to legitimate publication in Pharmaceuticals. As the authors mention in the conclusion section the work of the recent 5 years has expended the field in terms of therapeutic applications of these compounds enormously and more can be expected in the furture.

Well, the manuscript is concisely and intelligibly written. The references are sufficient and the reviewed papers are linked properly into several broader contexts by citing various review articles and works of similar fields.

To adress at least one little thing:
Though the self-citations are comprehensibly related to the context of the work I suggest to leave reference 26 out, since it is allready cited in the review article 29, which is in turn cited in the same passage of the manuscript.

Nonetheless I suggest to accept the manuscript in its present form.

Author Response

Response to reviewer 3:

Thank you for your valuable comments and suggestions. In a response to your criticism on citing several our previous papers, we have deleted the reference No. 26, as suggested.

Thank you for this positive review.

Reviewer 4 Report

In this manuscript entitled: Saponins of Selected Triterpenoids as Potential Therapeutic Agents: A Review, the authors: Uladzimir Bildziukevich, Martina Wimmerová, and Zdeněk Wimmer * present the review focused on saponins of selected triterpenoids as potential therapeutic agents.

I believe that the paper may be suitable for publication in the MDPI journal Pharmaceuticals after addressing the following considerations listed below.

- Figures should be placed in the main text near to the first time they are cited. For example, Figure 1 is cited in the main text in lines 28 and 29, therefore I suggest it be placed in line 38 and the title of Figure 1 in line 39. Also, at the end of the title of each figure, there needs to be a period.

- In line 52 (Figure 1), I suggest writing “1f, R = CHO” instead of “1f, R = -CH=O” and “1g, R = CH3” instead of “1g, R = -CH3”.

- It is necessary that the empty lines be deleted., e.g., lines 51 and 53.

- In lines 121, 174, and 437, I suggest that “μg.mL-1” be replaced by “μg · mL−1.

- I suggest that foreign words not be italicized, including Greek/Latin terms, such as in vivo (e.g., in line 256), in vitro (e.g., in line 404), and in silico (e.g., in line 135). 

- In line 236, I suggest that “mg.kg-1.day-1” be replaced with “mg · kg −1/dayor “mg · kg −1 per day or mg/kg body weight per day”.

- In line 243 (Figure 9), I suggest indicating some representative examples (R1 and R2) of the 56 plant products mentioned in reference 53.

- I suggest that line 252 be reformulated.

- In line 246, I suggest that “Platycosides of the general structures shown in the formulae 9a9c (Figure 9)” be reformulated, for example with “Platycosides having the general structures 9a9c shown in Figure 9” or “Platycosides with the general structures 9a9c (Figure 9)”.

- In line 376, after references, a period is missing.

- I suggest that do not leave a space before the percentage (%) symbol, e.g. in line 424, I suggest writing “80%” instead of “80 %”.

- I suggest that “cf.” be deleted or if it is necessary indicate the word instead of the abbreviation. For example, in line 140, I suggest writing “(Table 1)” instead of “(cf. Table 1).

Author Response

Response to reviewer 4:

Thank you for your valuable comments and suggestions. Our response to your criticism is summarized below:

Figures and Tables are always placed as close as possible to the text, in which they are mentioned. However, no Figure or Table is allowed on the first page of the paper (or the manuscript, in this case, as well). Therefore, Figure 1 is shown on page 2 of the manuscript immediately, when possible. The Figures 3-5 are quite big, and they are connected with a single paragraph. It is not easy to find where to place them and respect all requirements. Nevertheless, we tried to find a solution in constructing the final revised manuscript. I have an experience from the past that the Editorial office people make always final changes in the final manuscript, even after the manuscript is displayed on the web page of the journal.

Figure 1 was changed as suggested.

Deleting of the empty lines is usually made in the final manuscript. I have made it wherever possible.

Lines 121, 174 and 437: Corrected.

Latin words: Most of the journals accept or even suggest to use italics for Latin words and names. Because other reviewers did not criticize that system, we left italics for all Latin words and names.

Line 236: Corrected.

Line 252: Corrected.

Line 246: The sentence was re-formulated.

The space between the number and the “%” mark was deleted several times in the text. The abbreviation “cf.”, used several times in the text, was deleted as well.

Round 2

Reviewer 1 Report

Introduction: Authors need to highlight   the originality  of this review paper compared to their recent reviewer papers.

Author Response

Response to reviewer 1, review No. 2:

Thank you for your valuable comments and suggestions.

Reviewer remark:

Introduction: Authors need to highlight the originality of this review paper compared to their recent reviewer papers.

Answer: We have introduced a more detailed paragraph in the end of Introduction that highlights the importance and originality of this type of the review paper.
